# 3D-structured supports create complete data sets for electron crystallography

Julian T.C. Wennmacher[1,2], Christian Zaubitzer[3], Teng Li[2], Yeon Kyoung Bahk[4,5], Jing Wang [4,5], Jeroen A. van Bokhoven[1,2] & Tim Gruene [1,6]

3D electron crystallography has recently attracted much attention due to its complementarity to X-ray crystallography in determining the structure of compounds from submicrometre sized crystals. A big obstacle lies in obtaining complete data, required for accurate structure determination. Many crystals have a preferred orientation on conventional, flat sample supports. This systematically shades some part of the sample and prevents the collection of complete data, even when several data sets are combined. We introduce two types of three-dimensional sample supports that enable the collection of complete data sets. In the first approach the carbon layer forms coils on the sample support. The second approach is based on chaotic nylon fibres. Both types of grids disrupt the preferred orientation as we demonstrate with a well suited crystal type of MFI-type zeolites. The easy-to-obtain three-dimensional sample supports have different features, ensuring a broad spectrum of applications for these 3D support grids.

[1] Energy and Environment Research Division (ENE), Paul Scherrer Institut, CH-5232 Villigen, Switzerland. [2] Institute for Chemical and Bioengineering, ETH Zurich, CH-8093 Zurich, Switzerland. [3] Scientific Center for Optical and Electron Microscopy (ScopeM), ETH Zürich, CH-8093 Zürich, Switzerland. [4] Institute of Environmental Engineering, ETH Zürich, CH-8093 Zürich, Switzerland. [5] Laboratory for Advanced Analytical Technologies, Empa, CH-8600 Dübendorf, Switzerland. [6] Present address: Institute of Inorganic Chemistry, University of Vienna, AT-1090 Vienna, Austria. Correspondence and requests for materials should be addressed to T.G. (email: tim.gruene@univie.ac.at)

Many objects have a specific orientation when they lie on a flat surface. A flat box falls on one of its largest faces, and needles do not stand on their tip. Many imaging and diffraction techniques require three-dimensional access to a sample to provide accurate information about its structure. Preferred orientation of the sample often hinders complete three-dimensional access[1–7]. Methods based on electron radiation are particularly affected, because the electron beam cannot penetrate the sample support, which is generally much thicker than the sample and which can shade the sample from the electron beam. Several methods address this problem: in two-dimensional electron crystallography and electron tomography, the missing data are partially recovered by extrapolation of the phase and the amplitude[8,9]. Three-dimensional crystals can be shaped by ion beam milling to force a different orientation[10,11], or they are embedded in a resin, which changes their orientation[12,13]. Incomplete electron diffraction data can also be complemented with X-ray powder diffraction data[14]. In cases of high symmetry space groups, complete data can be collected from a single crystal. However, ~60% of all published crystal structures, both macromolecular and organic compounds, have a low-symmetry space group[15,16]. Every method that produces crystals in different orientations makes it possible to combine data from several crystals to reach complete data. Furthermore, when these orientations are random, averaging leads to improved data quality. Crystallography is particularly well suited for the combination of different data sets, because determination of the sample orientation is an essential part of data analysis. The correct combination of crystallographic data, called data merging, is well developed[17,18]. Many materials are transparent to X-rays, and diffraction data can be collected from a crystal in any orientation. In electron diffraction, the situation is very different. The strong interaction of electrons with matter, which enables the study of micrometre- and nanometre-sized crystals[5–7,19], prevents electrons from penetrating the sample supports. As sample supports have a flat surface, any crystal with a flat shape and a low symmetry space group will result in incomplete data, even when data from many crystals are merged[20–22]. A conventional sample support, used in transmission electron microscopy (TEM), is a thin, flat carbon film transparent for electrons. It is stabilised by a copper or gilded metal grid, which is opaque for electrons. This creates a dead zone with no data, known as the missing wedge problem[23,24]. The systematic lack of diffraction data leads to a distortion of the electrostatic potential map. This map is the basis of the structural model, and thus the missing wedge considerably reduces the reliability of the atom coordinates and of the atomic displacement parameters. Hence, although sometimes structural information from a single crystal may be desired, it is generally

more important to reach full data completeness at the expense of merging data from several crystals. An example where data completeness is crucial is the crystallographic study of aluminium substitution in the zeolite ZSM-5. Zeolites are microporous aluminosilicate minerals. The framework structure of ZSM-5 is well known and not the focus of crystallographic studies. Substitution of $Al^{3+}$ for $Si^{4+}$ in the siliceous framework generates a charge imbalance, which must be compensated by introducing non-framework species, such as protons and redox-active cations, giving zeolites their catalytic activity[25]. The location and precise speciation of such species requires high-quality data.

Here, we propose an effective and generally applicable method that solves the problem of incomplete data by introducing two types of sample support with a three-dimensional structure. One approach causes the transparent carbon film to form coils. Crystals stick to the curved surface in many different orientations. The second approach leads to coverage of the sample grid by thin nylon fibres with a diameter similar to the crystal size. This creates a mesh that leads to random orientations of the crystals. For both types of support we present 100% complete data from less than five crystals.

## Results

**Zeolite ZSM-5 as suitable sample with preferred orientation.** The copper support of conventional TEM grids restricts the maximum available rotation angle to ~150° when the sample is centred between the grid bars. The angle is further reduced when the crystal is off-centre (Fig. 1a, b).

We chose the zeolite ZSM-5 as specimen to improve the missing wedge problem. ZSM-5 is a zeolite of the MFI framework[26]. Careful synthesis of ZSM-5 yields crystals that are very well suited to study the missing wedge problem[27]. ZSM-5 crystals resemble a flat box. The direction of the short crystallographic **c**-axis coincides with the short macroscopic edge of the crystal. ZSM-5 crystals are, therefore, an ideal representative of crystals with a preferred orientation[28,29]. Chemical leaching of ZSM-5 leads to a cylindrical cavity that runs parallel to the crystallographic **c**-axis[30]. This provides electron-optical control of the preferred orientation of the crystallites. It shows up as a dark ring surrounding the entire crystal (Fig. 1c). The importance of data completeness is shown in Fig. 2. At atomic resolution and when the data are complete, the electrostatic potential map is composed of spherical blobs and the model atoms have a well defined position at the centre of every blob (Fig. 2a). Incomplete data causes a stretching of the blobs in the direction of the missing wedge. Even with only 10% data missing, the elongation of the map is visible (Fig. 2b). Typical data from 3D electron diffraction are only 70–80% complete (Fig. 2c, d).

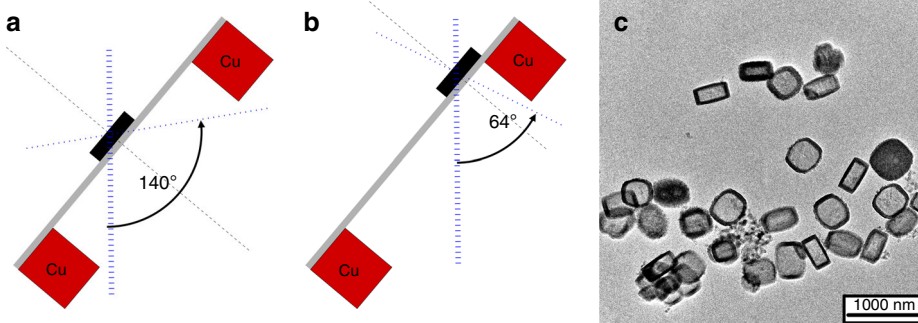

**Fig. 1** Restricted view in electron diffraction. Conventional sample support leads to incomplete data. **a** When the crystal is centred between grid bars, the maximum rotation range is ~140°. **b** A crystal close to the grid bar reduces the maximum rotation range. **c** TEM micrograph of ZSM-5 crystals illustrates their preferred orientation

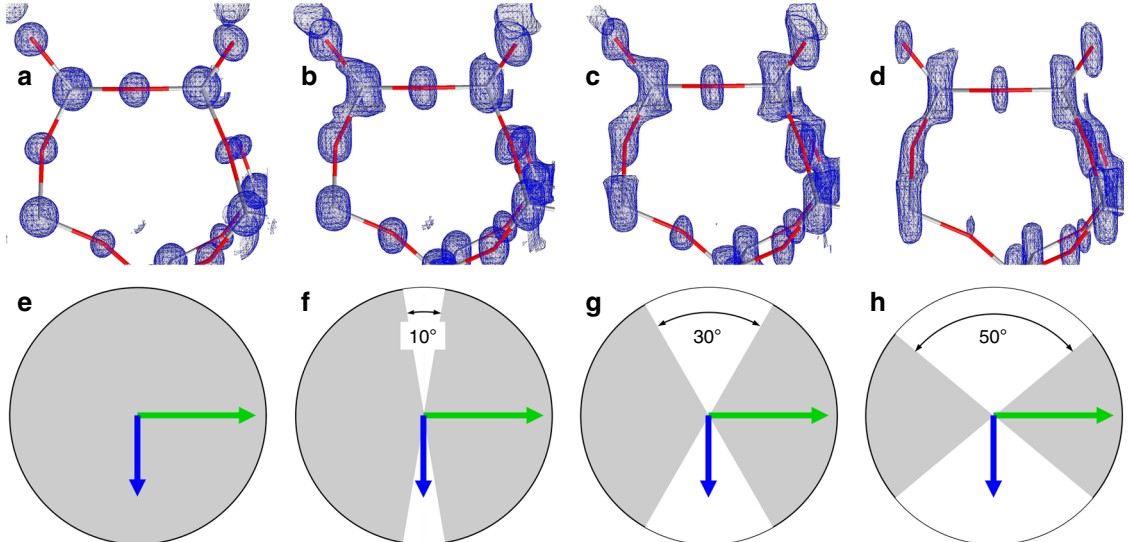

**Fig. 2** Importance of data completeness. Small section of the ZSM-5 framework (Grey-red bars: T–O bonds) shows that incomplete data lead to elongated electrostatic potential maps. **a** 100% complete data. **b** Missing wedge with 10° opening, 90% data completeness. **c** Missing wedge with 30° opening, 66% data completeness. **d** Missing wedge with 50° opening, 45% data completeness. **e–h** illustration of the missing volume in reciprocal space. Grey: observed data, white wedge: missing data. Green arrow: crystallographic **b** axis, blue arrow: **c** axis

**Coiled carbon film**. Leached, ring-shaped ZSM-5 crystals were deposited on a conventional, flat carbon support. Gentle striking of the carbon support with a fine-haired brush caused the carbon film to coil up (Fig. 3a). Crystals attached to the surface followed the curvature and, thus, changed their orientation. (Fig. 3a, b, and Supplementary Fig. 2). The number of coils per mesh depends on the applied pressure and can be monitored with a light microscope (Fig. 3c and Supplementary Fig. 1). We present data from nine randomly chosen crystals. Merging all nine data sets together yielded 100% data completeness. Subsequently, poorly fitting data sets were iteratively excluded from merging as long as this did not reduce data completeness. The poorest data set was selected based on the pairwise $CC_{1/2}$[31]. This way we were able to reduce the number of required data sets required to achieve 100% complete data to three (Fig. 3d, e).

**Nylon fibres**. Nylon fibres with a diameter of ~100 nm were deposited on commercial grids covered with lacey carbon film. This created a chaotic network with holes of various sizes (Fig. 4a–c). The density was controlled with the deposition duration and could be adjusted to the particle size of the investigated crystals: lower density resulted in larger holes. The size of the ZSM-5 crystals was between 400 and 600 nm, so the fibrous network and the zeolite crystals have similar dimensions. This is important in order to successfully break the preferred orientation of the crystal. A network based on nanometre-sized carbon tubes[32] is suitable for single particle analysis, but the network would appear flat relative to the size of sub-micrometre crystals. Figure 4d shows zeolite crystals entangled in the nylon fibre network. Judged only from this figure, the variation in the orientation is less obvious than with the coiled carbon film. Diffraction data were collected randomly from eight different crystals. The oscillation range for each crystal was ~120°. 100% complete data were obtained by merging diffraction data of four crystals (Fig. 4f, g). We did not notice an increased sensitivity of the grids or the nylon fibres to irradiation compared to conventional grids. However, when deposited on pure copper grids without a carbon layer, shifts in the fibres after strong irradiation became detectable (Supplementary Note 1 and Supplementary Fig. 4).

## Discussion

Imaging and diffraction techniques based on the transmission of electrons are impeded by incomplete data, as electrons do not penetrate the sample support. In some applications, particularly in crystallography, multiple data sets can be merged to obtain complete data. However, most crystals are flat and lie on the sample support in a preferred orientation. Therefore, merging of data from many crystals does not attain complete data. We describe here two different types of sample support, which lead to randomisation of the crystal orientation and, thus, to complete data from only very few crystallographic data sets. This way, 100% data completeness in electron diffraction studies is no longer restricted to crystals of high symmetry space groups, or to crystals with equant habit. Independent from our sample supports but owing to the early stage of development of electron diffractometers, the present study includes two data sets of very poor quality (see Supplementary Tables 1 and 3 and Supplementary Note 2). Without data set x10_5 for the data sets from coiled carbon film, and without the data set x11_11 from the nylon three-dimensional network, the respective sets miss 17 out of 4756 and 1 out of 4984 reflections, respectively, in the resolution range 1.2–1.0 Å. As fully integrated electron diffractometers are under current development in hardware and in software[33–35], such outliers can be compensated with data collection of additional samples, as has been common practice in X-ray crystallography. Both types of sample supports are easy to produce. One might suspect an increase of background noise. However, we did not observe such an increase, compared to our previous studies on ZSM-5 with conventional support grids[29]. We also did not observe any systematic deterioration in the scale factors (Supplementary Figs. 5 and 6). The two different types of sample support have different properties (Table 1) with advantages, depending on the application and sample properties. Coiling the carbon layer of the sample support is an easy and fast method. The concept could be further refined through the precise engineering of a sample support, e.g., based on graphene (Supplementary Fig. 3). Chemical modification of graphene may facilitate their use for this purpose and reduce potential diffraction from its regular structure[36]. 3D structured support grids based on nylon fibres are more reproducible, and they provide greater sample specific control: both the

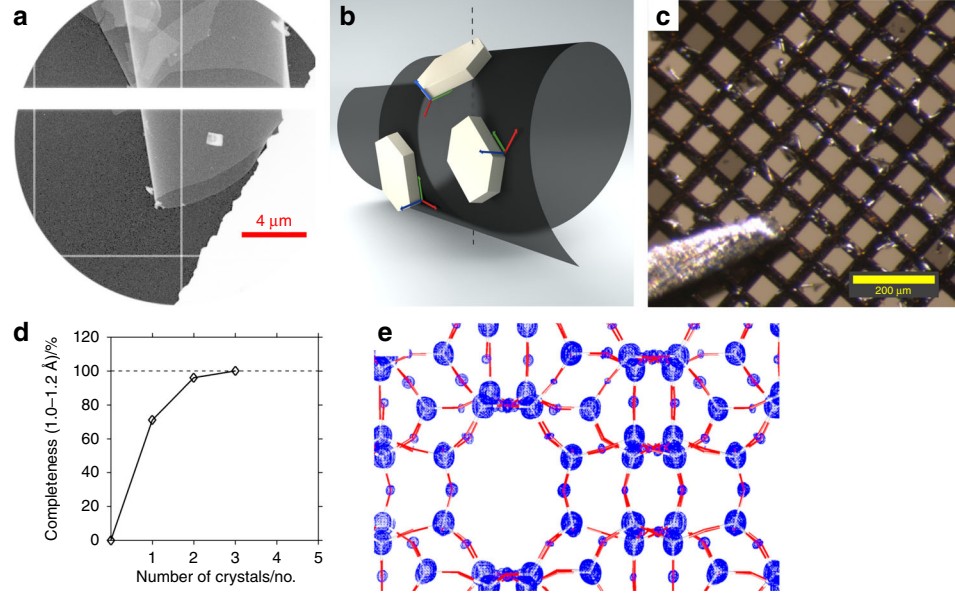

**Fig. 3** Complete data from coiled carbon film. **a** TEM micrograph of coiled carbon film with ZSM-5 crystal. More variations in the crystal orientations are shown in Supplementary Figs. 2 and 3. **b** Cartoon illustration of the randomised orientation of schematic crystals with flat shape attached to the coiled foil. **c** The coil is visible with a light microscope. **d** 100% data completeness is reached by merging three data sets. **e** Electrostatic potential map, calculated from 100% complete data from three crystals, results in reliable atom positions (Grey/red bars: T–O bonds of ZSM-5)

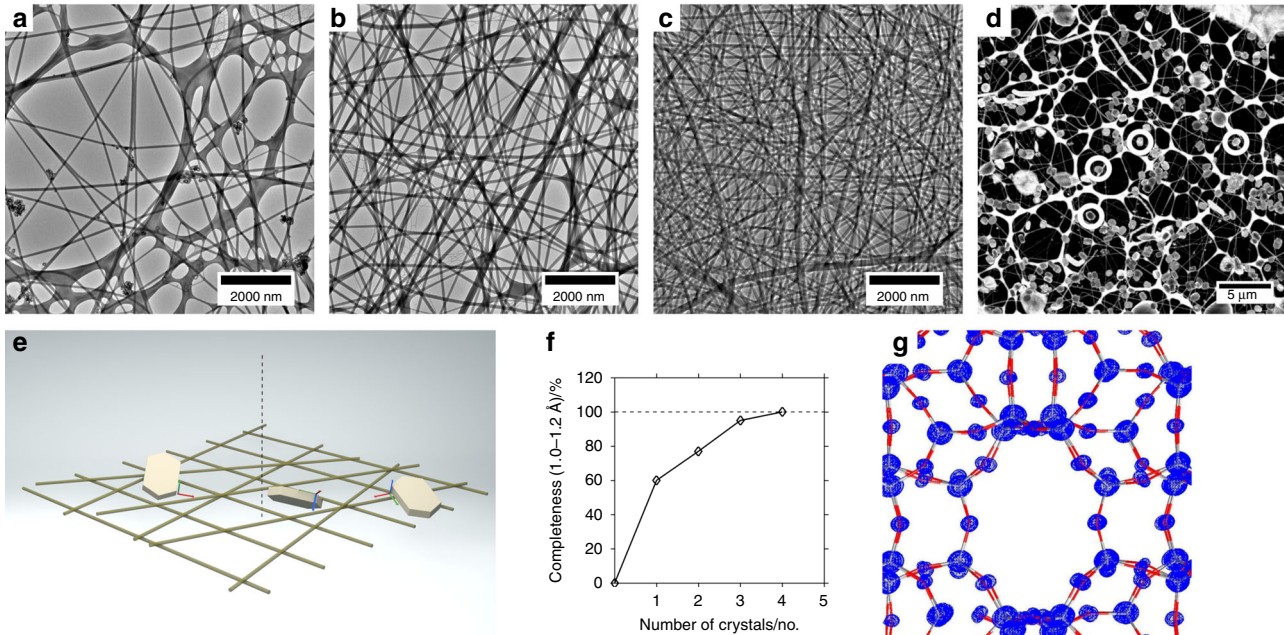

**Fig. 4** Complete data from nylon fibres. **a** TEM micrograph of a nylon-coated sample support with low fibre density. **b** TEM micrograph of a nylon-coated sample support with mid fibre density. **c** TEM micrograph of a nylon-coated sample support with high fibre density. **d** TEM micrograph of ZSM-5 crystals (encircled) entangled in nylon fibres. **e** Cartoon illustration of the randomised orientation of schematic crystals with flat shape attached to the nylon fibres. **f** 100% data completeness is reached by merging four data sets. **g** Electrostatic potential map, calculated from 100% complete data from four crystals results in reliable atom positions (Grey/red bars: T–O bonds of ZSM-5)

---

**Table 1 Properties of the two types of three-dimensional sample supports**

| Sample support | Preparation | Reproducibility | Dep. liq. | Bgd noise |
|---|---|---|---|---|
| 1. Coiled carbon film | + | − | − | + |
| 2. Nylon 3D-Network | + | + | + | − |

*Dep. liq.* support suitable for sample deposition from liquid suspension, *Bgd. noise* possible increase of background noise

diameter of the fibre and the density of the fibre network can be adjusted to the specimen. Nylon fibres are suitable for sample deposition in a liquid environment and can function as a sieve, e.g., for the study of protein crystals in combination with vitrification. Both types of grids are compatible with automated data collection[33]. Tomography can also benefit from the averaging of multiple data sets, and these grids may find applications outside the field of crystallography[37].

## Methods

**Preparation of MFI crystals**. The commercial TPAOH solution was purchased from Acros (25 wt% in water).

Tetrapropylammonium bromide (TPABr) was purchased from Fluka (≥98.0 wt%), sodium hydroxide from Merck (ACS reag. Ph Eur) and aluminium nitrate nonahydrate (Al(NO$_3$)$_3$ · 9H$_2$O) from

Acros (> 99 wt%). In a typical synthesis of ZSM-5 crystals, 12 g commercial TPAOH solution were added to a teflon reactor containing 12.5 g tetraethyl orthosilicate. The mixture was heated to 80 °C and stirred for 24 h at 500 rpm. After cooling down to room temperature, a solution of sodium hydroxide (0.24 g), aluminium nitrate nonahydrate (0.46 g) and deionized water (4 g) was added dropwise while stirring vigorously. The final gel composition was 1 Al$_2$O$_3$: 100 SiO$_2$: 25 TPAOH: 5 Na$_2$O: 830 H$_2$O. After homogenisation, the mixture was transferred to a 50 ml stainless steel autoclave equipped with PEEK inlets and heated to 170 °C for 24 h under static conditions. The product was separated by centrifugation for 15 min at 15,000 rpm, washed three times, dried overnight at 100 °C and calcined for 10 h at 550 °C. The lab-made template solution was prepared by stirring a mixture of TPABr (0.9820 $g$), commercial TPAOH solution (9 $g$) and deionized water (2.25 $g$) at room temperature[27]. Crystals were leached in NaOH solution (0.15 M, 35 ml per g zeolite) at 80 °C for 10 h. The product was separated by centrifugation for 15 min at 15,000 rpm, washed three times and dried overnight at 100 °C.

**Preparation of nylon 3D-network**. Nylon fibres on copper TEM grids (Lacey F/C 400 mesh Cu, Ted Pella Inc., USA) were prepared by electrospinning. Pellets with 15 wt% of Nylon-6 (particle size 3 mm, Sigma Aldrich, USA) dissolved in formic acid (purum ≥ 98.0%, Fluka, Germany). High voltage (20 kV) was applied to the solution through a stainless steel needle on a plastic syringe. A syringe pump (NE-300, New Era Pump Systems Inc., USA) constantly fed the solution to the electrospinning system at 1.5 μl min$^{-1}$. The TEM grid was placed on a stainless steel collecting plate. Electro-spun nylon fibres were deposited on the grid for 2 min to produce the network density shown in Fig. 4a. Deposition time of 5 and 8 min resulted in the network densities of Fig. 4b, c, respectively. The prepared nylon fibres grids were dried under the ambient condition for 24 h[38].

**Sample deposition**. The ZSM-5 crystals were applied either as a dispersion (1.5 mg ml$^{-1}$ EtOH) or as a dust cloud generated by a paintbrush on a commercially available continuous carbon TEM grid (Ted Pella, 01843-F), respectively, the nylon grids. In the former case, leached, ring-shaped crystals were used, in the latter case, solid, unleached ZSM-5 crystals were used. The continuous carbon TEM grids had been plasma-treated before. A fine-haired paintbrush was used cause the coiling of the continuous carbon film on the grids. The process is visible with the bare eye or with a light microscope (Supplementary Fig. 1). The grids were mounted so that the diagonal of the mesh was perpendicular to the rotation axis of the TEM sample holder.

**Data acquisition and processing**. Data were acquired and processed as described in[7,35]. In brief, data were collected at room temperature on a Tecnai F30 TEM (FEI, now ThermoFisher) equipped with a Schottky Emitter at an energy $E = 200$ keV, corresponding to the wavelength $\lambda = 0.02508$ Å. Diffraction data were collected in TEM bright field mode with a dose rate of ~0.01 e$^-$ Å$^{-2}$ s$^{-1}$ (The reading of the dose was displayed as 0.00–0.01 e$^-$ Å$^{-2}$ s$^{-1}$ with a precision of only two digits). Data were recorded with an EIGER X 1M detector (DECTRIS Ltd.). Data were processed with XDS[39]. To control the correct assignment of the **a** and **b** axes all data sets were solved with direct methods with SHELXT[40] in space group $Pnma$. Consistent indexing with respect to the **a** and **b** axes was controlled via statistics of the systematic absences[29]. Data statistics with respect to completeness refer to Laue group $\bar{1}$ (Supplementary Tables 1–4).

## Data availability

All (diffraction) data and respective CIF files are available at https://doi.org/10.5281/zenodo.2553377.

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

## Acknowledgements

We are grateful to DECTRIS Ltd. for lending us an EIGER X 1M and for technical support, in particular, S. De Carlo, C. Schulze-Briese, B. Luethi, D. Mayani, L. Piazza, and L. Wagner. We acknowledge N. Blanc, from ScopeM at ETH Zurich, for generous support and access to the FEI F30 TEM. J.T.C.W. thanks the PSI EM facility, in particular, E. Müller and T. Ishikawa, for their daily support and discussions. We thank M. Schoenberg for proofreading of the manuscript. We thank M. Dzambegovic for the cartoon illustrations. T.G. acknowledges the extraordinary trust of Chr. Schönenberger, SNI, for this project. T.L. thanks the China Scholarship Council (CSC) for financial support. Funding for this research was provided by the Swiss Nanoscience Institute (grant No. A12.01 A3EDPI). J.T.C.W. was supported by the Swiss National Science Foundation (project No. 200021_169258). T.L. was supported by the China Scholarship Council.

## Author contributions

Each author contributed to the manuscript. J.T.C.W. carried out the experiments, analyzed the data, and wrote the initial manuscript. C.Z. carried out experiments. T.L. synthesized the ZSM-5 sample. Y.K.B. provided sample supports. J.W. provided sample supports and resources. J.A.v.B provided ZSM-5 sample and resources. T.G. designed the experiments, analyzed and curated data, was responsible for main funding acquisition, conceived the method, supervised the experiments, and managed the project and the manuscript.

## Additional information

**Competing interests:** T.G., J.T.C.W. and J.Av.B. filed patent No. EP 18 202 868. The remaining authors declare no competing interests.

