## [Peer Review File · Nature Communications]

Reviewers' comments:

Reviewer #1 (Remarks to the Author):

This communication gives very good suggestions in order to get complete diffraction data set from nanocrystal by using MFI as an example.

It will be nice to discuss following points in the text;

(i) If we will have a complete diffraction data set from one nanocrystal, that will be perfect. However depending on crystal symmetry, necessary volume in reciprocal space for collecting diffraction intensities will be reduced and therefore seriousness introduced with missing-cone problem depends on crystal symmetry of the structure solution.

(ii) This approach request four nanocrystals of MFI to reach 100 % completeness (Figure 3). This means structure solution will be an averaged solution over four nanocrystals, although we want sometime to obtain the solution only from one nano crystal.

(iii) Both carbon film and nylon fiber seem to be electron beam (e-beam) sensitive. It is therefore important to show with experimental evidences that they are mechanically stable enough during observation with e-beam.

Reviewer #2 (Remarks to the Author):

A common problem in TEM sample preparation is that crystals/particles lie with preferred orientations on the TEM grid. The preferred orientation causes problems in collecting complete 3D data, which is required for structure analysis. In the current manuscript, the authors report two simple and effective approaches to overcome such a problem by creating 3D-structured supports on TEM grids. One approach is to generate coiled carbon support from TEM grids coated with thin flat carbon films. The second approach is to deposit nylon fibres on TEM grids to create a structured network of nylon fibres. Using zeolite ZSM-5 crystals as a test example, the authors show that the 3D-structured supports on TEM grids can suppress the preferred orientation of ZSM-5 crystals. Complete 3D electron diffraction data sets can be obtained by merging data from a few crystals and used for structure determination. These novel and innovative approaches are of general interests in

several fields from chemistry, materials sciences to life sciences. I recommend the publication of the manuscript after the following comments have been addressed.

1. The author pointed out that it is important to match the dimension of the fibrous network on with the crystal size. How about the diameter of the carbon coils? Is it tunable? The diameter should also play an important role. This should be addressed in the manuscript.

2. How is the stability of the coiled carbon film and the nylon fibre under electron radiation? Any changes in the support may cause orientation changes during data collection, and therefore affect the data quality. This is important and should be addressed.

3. According to the cif file of the ZSM-5 structure refined from the data obtained using coiled carbon support, there are many unreasonable bond distances and angles, for example

Si2 O7 1.19(5)

Si3 O9 2.17(7)

Si4 O9 1.23(7)

O7 Si2 O6 135(3)

O7 Si2 O5 91(3)

O9 Si4 O30 87(3)

O28 Si14 O31 78(2)

The ideal Si-O distance is 1.61 Å and the O-Si-O angle 109 degrees. I suggest the authors to introduce distance restraints in the refinement.

Some minor typos:

4. Page 5, line 99, "Both types of grids a compatible" should be "Both types of grids are compatible"

Table S4: delete the first number "1" in the cell 1 19.955(7).

Reviewer #3 (Remarks to the Author):

The manuscript describes two ways how it is possible to break preferred orientation of the crystals, which is often observed in electron diffraction experiments in transmission electron microscopes. The first is based on a controlled breaching of the thin amorphous carbon foil using paintbrush and the other is a deposition of nylon fibres on standard TEM grid, which introduce surfaces not coplanar

with the carbon foil. Both of these methods are novel and in my opinion they may help to fight the preferred orientation problem. The presented solutions will probably appeal to community of electron crystallographers. I cannot judge how it will appeal to the broader audience. I tried to imagine myself using these techniques to fight the preferred orientation and I am still not persuaded that I would use it. It seems from the Fig. 3c that with the method of coiled carbon foil you more or less destroy the foil and finding a crystal suitable for the measurement could be quite tedious. The preparation of the nylon covered grid is relatively specialized (people doing electron crystallography do not have electrospinning setups). I think I would rather use lacey-carbon grid and spend an extra hour to find the crystal in a hole.

I see the potential major problem with these techniques in the limited transparency of the materials to the electron beam and I would like the authors to carefully comment this issue. In the case of materials composed of only light elements (2nd row of periodic table) acquiring of sufficiently good data from crystals thicker than 100 nm is difficult. The crystals, which do have the preferred orientation problem are mainly platelets which are only nano in 1D like in the case of the presented zeolite crystals. If you manage to turn the crystal out of its preferred orientation thanks to the 3D support how can you measure sufficiently good data if the effective crystal thickness increases to hundreds of nanometers? In my opinion, the resolution of the diffraction data acquired from crystals tilted from their preferred orientation by angles more than 75° will show significant decrease of resolution. If one looks in the Tables S1-S3, the I/σ values are frequently below accepted limit of 2 for the highest resolution shell (usually about 1 Å) - this means there are virtually no data. Are the data acquired from highly tilted platelets suitable for structure solution (i. e. do they have sufficient resolution)? This is not clear from the paper.

Minor things:

1) I did not understand how graphene could be used as a support for electron diffraction experiment as it is crystalline and so it would produce diffraction maxima.

2) The authors state that the reproducibility of coiled carbon support is problematic. My opinion is similar. The quality of the carbon foil may be different in different production batches. Did the authors try also other TEM grid manufacturers to confirm that the carbon foils can be coiled by the paintbrush method and so that the method is generally applicable?

3) Lacey-carbon foils represent a relatively good support to fight the preferred orientation - you may find crystals, which are hanging on the hole edges (partially or completely). Could the authors describe the advantage of their approaches in comparison to the use of commercially available and easy-to-use lacey-carbon TEM grids?

Mistypes:

manuscript line 99 "are" instead "a"

suppl. line 6 remove "be"

Two aspects were specifically highlighted, the stability of the supports and a potential loss of resolution, which we would like to address first.

Stability:

We provide data represented in Supplement Fig. S4, referenced on p.7, l.94. The top row shows two images of the same sample, 30s apart. The difference of both figures (Fig. S4-C) shows only noise, i.e. there is no difference between both time points. Fig S4 D-F present the positive control: the difference (S4-F) of two images S4-D and S4-E with a deliberate difference does show features. As shown in S4G-J, radiation damage occurs under radiation much stronger than we used for our experiments and(!) when there is no carbon layer.

Resolution:

Before this study, we collected many data sets from ZSM-5 with conventional grids (in the context of Gruene, Li et al (2018)). During data collection for the current manuscript, we did not observe obvious differences with respect to resolution. We cut the resolution at 1.0Å, because this is the "largest common resolution" taking into account that the beam was not always placed at the same position on the detector and that there are variations in crystal volume and quality. Most of these crystals diffracted beyond 0.7Å resolution (please note that the raw diffraction images will be published at zenodo.org, DOI 10.5281/zenodo.2553377 alongside with the manuscript). We provide additional images S5 and S6 in the Supplement. They show the scaling factors for each data set, binned across frame number and across resolution range. The variation in the scaling factors does not seem to show a trend.

Reviewer 1

=====

(i) We added to the introduction (p.1, l. 22), that structures with a reasonably high symmetry space group are not affected by the missing wedge. In order to place our work into this context, we also add a note about space group statistics in the CSD and PDB. In materials science, the distribution of space groups may be different, but also structural studies in materials sciences have often different objectives than for organic or macromolecular structures,

where incompleteness is much more of a problem due to radiation sensitivity of the samples.

(ii) We added to the introduction (p. 2, l. 43) that there are cases where data completeness is of lesser importance than solving the structure from a single specimen. In organic and macromolecular crystallography, this is rarely the case. A single crystal used for X-ray structure determination has a much greater volume than a few crystals used for electron structure determination. We could not find an example of a chemical structure where data completeness is less important than solving the structure from a single crystal, hence we could not provide a reference.

(iii): see above, general remarks on stability

Reviewer 2

=====

1. The effect of the coiled carbon is very different from that of the nylon grids. The coil describes a rotation of more than 180 degrees. A crystal randomly attached to the surface is going to have a random rotation between 0 and 180 degrees. In addition, the rotation axis of the goniometer will also have a random orientation with respect to the coiling axis. The strength of the rotation method lies in the random sampling of reciprocal space, and the coiled carbon adds to this strength. This effect does not depend much on the coiling radius. However, when the coiling radius is very much smaller than the crystals, they may not be disturbed, similarly to what we describe for the nylon fibres. The brush procedure, as depicted in Supplement Figures S1-S3, produces structures between several hundred nanometer and a few micrometer (Fig. S1c shows various colours, suggesting that the size features are similar to the wavelength of visible light (400-800nm)). Crystals much larger than micrometers are going to be opaque to electrons and thus anyway not suitable for data collection. For cases where the crystals are much smaller than the coiling radius, the effects as shown in Fig. S2 can be exploited. Since we did not explore this question systematically (and we do not have sufficient access to instrumentation, to do so thoroughly), we would prefer to leave those figures in the Supplement. During our experiments, the screening of the grids for suitable crystal (orientations) was not very time-consuming, as the features of the coils and the attached crystals are visible at low magnification of the

TEM.

However, as the resolution of 3D-printers becomes better and better, these may provide a future means for very fine-tuned control over the grid surface. Since at the current stage, however, this suggestion is only speculation, we would prefer to keep inside this response letter.

2. see above, general remarks on stability

3. Indeed we neglected the quality of the structure. We now scaled the unit cell parameters linearly so that the average Si-O bond distance is 1.609 for both CIF files (cf. Gruene, Li et al, 2018) and added the suggested bond distance restraints (DFIX 1.609). The angle O-Si-O angle has a very broad distribution (see eg. 10.1103/PhysRevB.67.014106). Therefore we consider it unsuitable as restraint in refinement. The variation in bond lengths is now reduced to 1.59-1.62Å for zsm5_coiled_mrg_publ.cif and 1.59-1.63 for zsm5_nylon_mrg_publ.cif respectively.

4. We corrected both typos as suggested.

Reviewer 3

The reviewer points at the production process of the nylon covered grids. We agree that commercial availability and industrial preparation with a variety of choices would be of great benefit, instead of making them by ourselves. The publication of our manuscript hopefully is a first step into this direction. On the other hand, YKB taught JTCW within 1-2 days how to prepare the grids, who has since prepared a few hundred copies. The required electrospinning apparatus can easily be set up with a syringe pump, a syringe, a rectifier, a metal plate, wires and a safety housing.

The reviewer remarks that data collection from crystals thicker than 100nm was difficult. We do not share this opinion. In our experience, thicker crystals can take a larger dose and the intensity of the reflections is proportional to the crystal volume. Both aspects act in the same direction and facilitate data collection from thicker crystals. The ZSM-5 crystals in our studies are far from 1D like, as shown in Fig. S2 and S3 and in greater detail in Li et al

(2018), which describes the synthesis of our samples.

There may be applications of electron diffraction, where the thickness of the sample is limited to 100 nm, but for chemical crystallography and related applications of the rotation method, literature does not reflect such a limitation. Sometimes, people also consider imaging as part of electron crystallography, where thickness may be indeed an issue, e.g. Zou/Hovmoeller, *Acta Cryst* (2008), A64, 149-160 or Mertin, *Angew. Chemie. Int. Ed.* (1997), 36, 46-47 (We acknowledge Frank Krumreich, ETH Zuerich, for pointing at this fact and these two references). Our manuscript refers to crystallography in the sense of structure determination from crystal diffraction data.

Minor aspects

(1) We incorporated the remark about diffraction of graphene into the manuscript and added a suitable reference, where the diffraction is disturbed by chemical modification. This makes the remark particularly interesting, as the chemical properties of graphene, its modifications and applications are active fields of research.

(2) We did not test other makers of grids. One of the strength of data collection with the rotation methods lies in the random orientation of the sample, and the non-reproducibility of the coiled carbon support adds to this strength. See also response to Reviewer 2, comment 1.

(3) The reviewer suggests that crystals hanging on the hole edges of lacey carbon to collect complete data sets.

In our study on the morphology of

ZSM-5, Gruene et al., *Chem Eur J* (2018), 24, 2384-2388, we used lacey carbon

grids (cf. Fig. 1c *ibidem*). However, even when merging all 10 published datasets (cf.

Table 1 *ibidem*), completeness reaches only about 97%. Hence, the reviewer might

be right that lacey carbon does help to increase resolution. However, we note

that even with 10 data sets, 100% was not achieved in our case. The advantages

of our approach:

- with coiled carbon, it is very easy to select different orientations of the

crystal, as the orientation varies with the carbon layer, which is often easy

to see in the TEM.

- also with nylon fibres, a systematic search is facilitated in our opinion,

guided by the topology. E.g. the crystal in Supplementary Fig. S4-A is

supported by a single nylon fibre close to its centre and therefore would

be

inclined as much as possible

- the main advantage of the nylon fibres, however: they should also work with

samples in solution, as mentioned in Table 1.

Both typos were corrected.

Reviewers' comments:

Reviewer #1 (Remarks to the Author):

The two methods newly proposed are very powerful and the methods and detailed techniques are well described.

The data completeness is ~ 100 % will be ideal as discussed and shown in this paper.

This will give a big value for scientists who are struggling to handle with sheet crystals, so I am for accepting this for publication.

However, I want to make two comments;

(1) In the case of zeolites, we can obtain framework structure even at missing wedge with 30 degree open and 66 % data completeness (corresponding to Fig 2c). So, for the both cases one crystal data is basically enough.

(2) ZSM-5 crystal is not a good example to show the power/advantages of these two approaches, as the crystal doesn't show severe "preferred orientation effect". It is better to take an example from sheet crystals like clays.

Reviewer #2 (Remarks to the Author):

The revised manuscript has satisfactorily addressed all my comments. I am happy to recommend it for publication in Nature Communications.

Reviewer #3 (Remarks to the Author):

The authors improved the manuscript. But there are two things which are still not solved in my opinion.

1) Table S5 shows how the scale behaves as a function of resolution but it does not show how it behaves as a function of the angle between normal to the platelet and the beam - this would be really informative because this is what the authors should want to show to the audience - look how we fight with the preferred orientation and how our data are independent on the effective thickness of the platelet when we tilt it. You write that your data are of sufficient quality even if you shining through the platelet - binning 130° long dataset in one to four columns does not persuade me about this. Of course you should take care of the changing cross-section of the platelet if you want to be in the absolute scale. I suggest also to put the angle between normal to the platelet and the beam in Table S1.

2) Regretably, I am not satisfied with the answer concerning the resolution - Table S1 shows I/σ below acceptable limit of 2 for four crystals out of six but you write that your crystals diffract to resolution 0.7 Å. So where is the problem? You have wrong noise parameters (sigmas), or you cannot integrate properly your high diffraction angle reflections because the crystals are not stable on the support (or the support itself is not stable), or something else? Look at your Rmeas values - they are below 40% nearly only for three dimensional network supports and datasets with I/σ more than two. For carbon films you have (Table S1) for the resolution shell ~ 1.6 to ~ 0.7 Å Rmeas about 100% (twice more than 200%). This indicates that your data have very large distribution of measured intensities for a particular reflection and the same problem persists in the merged data from several crystals - Tables S2 and S4. I am sorry, but I doubt that the quality of the data produced by these innovative approaches are of sufficient quality for general material structure solution - zeolites have very uneven distribution of scatterers, which substantially helps in structure solution. But in materials without voids/pores composed of atoms with similar scattering power I would expect that Rmeas over 40% would mean that you will not be able to solve the structure unless you combine tens of datasets. And I have to say that such a large Rmeas cannot be explained by dynamical effects. I feel that the authors did not addressed the data quality issue properly in the manuscript.

We incorporate the reviewers' comments in the manuscript without modification of the data, but by extending the introduction and the discussion section.

Reviewer 1 remarks that for ZSM-5 - the sample that we used in our study - complete data are not necessary to solve the framework structure of this zeolite. This is correct, and the framework is not the information we are looking for. We added one paragraph to the introduction (p. 2, l. 59+) that states that the framework structure is not the focus of interest and that explains why we need complete high-quality data for our study for the questions that we actually want to address concerning ZSM-5, namely indirect evidence to the Al-substitution with determine the catalytic of ZSM-5.

Reviewer 1 also remarks that ZSM-5 was not well suited for a study about preferred orientation, "as the crystal doesn't show severe preferred orientation effects". Reviewer 1 may have commercial ZSM-5 in mind - there seem samples where the crystals resemble a cauliflower like shape. The manuscript now stresses more strongly than before that our ZSM-5 crystals are specifically synthesized according to the cited literature ([27], Li et al, ChemNanoMat (2018)). We modified the sentence on p. 3, l. 78 to "Careful synthesis of ZSM-5 yields crystals that are very well suited to study the missing wedge problem." with a reference to the respective protocol. Our crystals are shown in Fig. 1c, and their special habit is described in the manuscript "It shows up as a dark ring surrounding the entire crystal (Fig. 1c)" - p. 3, l. 84

Reviewer 2 is "happy to recommend [the manuscript] for publication in Nature Communications.", hence requires no further discussion in the manuscript.

Reviewer 3, remark (2):

Reviewer 3 discusses the resolution cut-off and refers to the I/sigma-rule as cut-off. We cut the data according to the 0.1% significance level of CC1/2 and cite the corresponding paper by Karplus and Diederichs (Science, 2012) in Tables S1 and S3. In the Supplement, Section B, we discuss the data quality with reference to Rupp's recent discussion of quality indicators (Structure, 2018).

We also added the comment in the Supplement that most of the rules for resolution cut-off in crystallography are valid with complete data and a high redundancy. They become invalid when data are incomplete. This can be conveniently tested by generating the statistics tables (e.g. with XPREP) after

reducing data completeness. However, this discussion goes far beyond the scope of our manuscript, and also is rather independent of the 3D support grids, but inherent to electron diffraction data collected with the rotation method and variations thereof. We therefore refer to practically all structural studies based on electron diffraction and based on the rotation method of data collection. Many of these publications (including our own preceding studies) show data quality worse than our data - note that some of these publications do not even display the statistics typically presented with crystallographic studies). Please also note that while we present the statistics up to high resolution, up to 0.7Å, the graphs in Fig. 3 and Fig. 4 are based on a conservation resolution cut-off at 1.0Å, as shown in the label for the Y-axis.

However, two data sets do not meet the 0.1% significance level of CC1/2. As discussed with the editor, we did not exclude these two data sets from the manuscript. Instead, we briefly discuss these outliers in the context of current developments of electron diffractometers, that will enable convenient and faster data collection at better quality and therefore make it easy to replace such outliers with better quality data.

Reviewer 3, remark (1)

Reviewer 3 also refers to the scaling factors and the variation of data quality as the crystal rotates. The reviewer suggests to plot the scaling factor as a function of rotation angle. This would be possible with very high inaccuracy, because with our work-around electron diffractometer (described in [35] Heidler et al, Acta Cryst D (2019)), the rotation and data collection are started independently and manually with an unknown delay, so that the exact angle of rotation is not known. But even so, the variation that reviewer 3 refers to, is inherent to the rotation method, and not inherent to our 3D structured support grids. Our grids merely overcome the main limitation in 3D electron diffraction based on the rotation method, namely the fact the data cannot be collected from a full rotation of a single crystal. Since our manuscript does not discuss the rotation method per se (this is done by many papers before, some of which we cite), we feel that a discussion of the variation of data quality with changing angle of incidence, that goes deeper than Fig. S5 and S6 is too much out of

scope of our work and would rather distract. As also mentioned our response to reviewer's 3 remark (2), we do cite a selection of representative papers that show data quality as good (or worse) than our data. In order to address this aspect in the manuscript, we refer to our previous study with ZSM-5 ([29] Gruene et al., Chem Eur. J. (2018)) which exhibits rather similar data statistics (please note that data quality varies from crystal to crystal even when collected under identical conditions).

REVIEWERS' COMMENTS:

Reviewer #3 (Remarks to the Author):

The manuscript was improved and the fundamental problem I had with the data quality measured from direction perpendicular to the normal to the largest facet of the crystals was clarified by Figures 4a and S4a/b.

In the original manuscript, the shape of the crystals was described as “ZSM-5 crystals resemble a flat box”. I read the information in a closely following sentence „Chemical leaching of ZSM-5 leads to a cylindrical cavity that runs parallel to the crystallographic c-axis [26].“ as an interesting feature of this material and not that this procedure was actually used for the alteration of the „flat box“ shape to an annular one. Therefore, I did not believe that it is possible to obtain good quality data when shining through the orientation where the crystal cross-section is the largest.

My first remark to the revised manuscript was not understood correctly. The scale should be depicted “as a function of the angle between normal to the platelet and the beam”, thus you should use your orientation matrix and not the actual rotation angle. Never mind, with annular shape of the crystal, I do not believe that the scale would be more or less proportional to the relative illuminated area of the annulus.

I admit that the sample chemical leaching was clearly mentioned in the first revision of the manuscript and that Figure S4a/b was also present in the revised SI. However, the Figures 3b and 4e are misleading, because they depict the crystals as platelets or boxes if you want. Also the “flat box” crystal shape description remained, even though the crystals have large cylindrical void leached by NaOH.

My recommendations and remarks:

- 1) Clearly state in the manuscript that the crystal shape is annular and if possible redraw the Figures 3b and 4e where are shown the schematic crystal shapes to reflect the actual shape of the crystals.
- 2) The applicability of these two methods for fighting with the preferred orientation was showed for the annular crystals, thus a rare morphology. In my opinion, the applicability for real platelets remains doubtful.
- 3) I went through the references 29 (you suggested) and 33 and 34 and there were never Rmeas values higher than ~100%, while you have in this manuscript two datasets with Rmeas more than 200%.

I think that this manuscript contains interesting approach to the treatment of the preferred orientation but I am not persuaded that general applicability of these methods was delivered. I leave it to the editor if this work is sufficiently interesting for a general reader of Nature Communications or if it would be more appropriate for a more specialized magazine.

The issue from Reviewer #3:

"1) Clearly state in the manuscript that the crystal shape is annular and if possible redraw the Figures 3b and 4e where are shown the schematic crystal shapes to reflect the actual shape of the crystals."

a) We added "Leached, ring shaped" in the results section of the 'Coiled Carbon Film"

b) We made captions for Fig. 3b and 4b more general, "Cartoon illustration of the randomised orientation of schematic crystals with flat shape [...]", i.e. replaced 'ZSM-5 crystals' with "schematic crystals with flat shape"

c) We added to the Method section the sentence " In the former case, leached, ring-shaped crystals were used, in the latter case, solid, unleached ZSM-5 crystals were used."